# Abstinence from Escalation of Cocaine Intake Changes the microRNA Landscape in the Cortico-Accumbal Pathway

**DOI:** 10.3390/biomedicines11051368

**Published:** 2023-05-05

**Authors:** Vidhya Kumaresan, Yolpanhchana Lim, Poorva Juneja, Allison E. Tipton, Giordano de Guglielmo, Lieselot L. G. Carrette, Marsida Kallupi, Lisa Maturin, Ying Liu, Olivier George, Huiping Zhang

**Affiliations:** 1Department of Pharmacology, Physiology & Biophysics, Boston University Chobanian and Avedisian School of Medicine, Boston, MA 02118, USA; aetipton@bu.edu; 2Department of Psychiatry, Boston University Chobanian and Avedisian School of Medicine, Boston, MA 02118, USA; ylim1@bu.edu (Y.L.); poorvaj@bu.edu (P.J.); d201077400@alumni.hust.edu.cn (Y.L.); 3Department of Medicine (Biomedical Genetics), Boston University Chobanian and Avedisian School of Medicine, Boston, MA 02118, USA; 4Department of Psychiatry, University of California San Diego, La Jolla, CA 92093, USA; gdeguglielmo@health.ucsd.edu (G.d.G.); lcarrette@health.ucsd.edu (L.L.G.C.); mkallupi@health.ucsd.edu (M.K.); lmaturin@health.ucsd.edu (L.M.); olgeorge@health.ucsd.edu (O.G.)

**Keywords:** protracted abstinence, outbred heterogeneous stock rats, prefrontal cortex, nucleus accumbens, rat brain miRNA, small RNA sequencing, addiction index

## Abstract

Cocaine administration alters the microRNA (miRNA) landscape in the cortico-accumbal pathway. These changes in miRNA can play a major role in the posttranscriptional regulation of gene expression during withdrawal. This study aimed to investigate the changes in microRNA expression in the cortico-accumbal pathway during acute withdrawal and protracted abstinence following escalated cocaine intake. Small RNA sequencing (sRNA-seq) was used to profile miRNA transcriptomic changes in the cortico-accumbal pathway [infralimbic- and prelimbic-prefrontal cortex (IL and PL) and nucleus accumbens (NAc)] of rats with extended access to cocaine self-administration followed by an 18-h withdrawal or a 4-week abstinence. An 18-h withdrawal led to differential expression (fold-change > 1.5 and *p* < 0.05) of 21 miRNAs in the IL, 18 miRNAs in the PL, and two miRNAs in the NAc. The mRNAs potentially targeted by these miRNAs were enriched in the following pathways: *gap junctions*, *neurotrophin signaling*, *MAPK signaling*, and *cocaine addiction*. Moreover, a 4-week abstinence led to differential expression (fold-change > 1.5 and *p* < 0.05) of 23 miRNAs in the IL, seven in the PL, and five miRNAs in the NAc. The mRNAs potentially targeted by these miRNAs were enriched in pathways including *gap junctions*, *cocaine addiction*, *MAPK signaling*, *glutamatergic synapse*, *morphine addiction*, and *amphetamine addiction*. Additionally, the expression levels of several miRNAs differentially expressed in either the IL or the NAc were significantly correlated with addiction behaviors. Our findings highlight the impact of acute and protracted abstinence from escalated cocaine intake on miRNA expression in the cortico-accumbal pathway, a key circuit in addiction, and suggest developing novel biomarkers and therapeutic approaches to prevent relapse by targeting abstinence-associated miRNAs and their regulated mRNAs.

## 1. Introduction

Recent epidemiological data suggest a resurgence in cocaine use and cocaine-related problems in the United States [1]. The continued use of cocaine induces lasting maladaptive plasticity that can lead to cocaine use disorder (CUD). Similar to other substance use disorders (SUDs), CUD is a chronic and relapsing neuropsychiatric disorder, causing a serious global health problem that exerts a heavy toll on both individuals and society [2,3]. CUD subjects are beset by a compulsion to use drugs despite severe, life-changing, and negative consequences. Drug overdose deaths involving psychostimulants, such as cocaine, are continuing to increase [4]. To date, no FDA-approved pharmacotherapies for CUD are available. Currently available medications for CUD are either maintenance or replacement therapies that are merely less addictive substitutes and do not address the true underlying pathology [5]. The identification of novel therapeutic targets, such as miRNA or their targets, and the development of treatment strategies based on miRNA, such as CUD-associated biomarkers, would greatly benefit CUD patients.

Evidence from family, adoption, and twin studies strongly suggests a genetic contribution to the abuse of alcohol and illicit drugs, including cocaine. Several genome-wide association studies (GWAS) have identified CUD-associated genetic variants; however, there is a lack of replicated findings [6]. Moreover, CUD-associated common variants identified so far by GWAS can only explain a small proportion of its genetic liability [7,8]. To uncover the molecular components that underlie the “missing heritability” of CUD, it is necessary to investigate the epigenetic mechanisms of CUD and gene–environment interactions, besides identifying CUD-associated rare variants that may exert a larger impact than common variants on CUD risk.

Aberrant epigenetic neuroadaptations induced by repeated drug exposure followed by protracted abstinence play a major role in the maladaptive plasticity of the cortico-accumbal pathways, leading to drug use disorders. Neuroadaptive changes can manifest as personality traits that render an individual vulnerable to neuropsychiatric disorders, particularly drug misuse. Epigenetic changes play an important role in behavior, health and disease. They are diverse, ranging from histone modifications, DNA methylation, noncoding RNA regulation, and chromatin remodeling [9,10].

miRNAs are ubiquitously expressed and play a major and crucial role in the posttranscriptional regulation of gene expression. miRNAs affect a wide range of biological processes in health and affect diseases including cancer and cardiovascular dysfunctions, among others. They play a critical role in CNS disorders, including cognition, neurodegeneration, and neuropsychiatric disorders [11,12,13]. miRNAs are noncoding RNAs of ~20–22 nucleotides that are negative regulators of protein translation. Each miRNA can target 100–1000 different mRNAs either for degradation or for repression of mRNA translation by binding to the 3′UTR of mRNAs. miRNAs thus regulate gene expression post-transcriptionally in response to external and internal stimuli in a neuronal activity-dependent manner [14,15,16]. miRNAs affect multiple aspects of neuronal and glial genesis, development, differentiation, and function, in particular dopamine neurons [17,18], as well as neuronal plasticity [19], cognition, and dementia [13]. Alterations in the miRNA expression landscape have been identified as causal events in co-morbid disorders such as stress and anxiety as well as substance abuse [16,20,21]. Therefore, targeting miRNAs and the mRNAs that they regulate is likely to be effective in preventing or reversing aberrant adaptations induced by chronic and compulsive drug use. Consequently, miRNAs may serve as novel therapeutic targets and as biomarkers of SUDs [21,22,23].

Cocaine-induced changes in brain miRNA transcriptomes constitute a major class of epigenetic alterations. Profiling of transcriptome-wide miRNA expression changes in the brains of CUD subjects facilitates the understanding of the epigenetic mechanism of CUD. Several rodent models are available that have face and predictive validity and mimic various core features exhibited by individuals with SUDs [24,25]. We have used N/NIH Heterogeneous Stock (HS) rats in this study. HS rats are unique in that they are highly recombinant animals; thus, they mimic the genetic diversity observed in human subjects [26,27]. The genetic diversity of HS rats ensures that some HS rats are vulnerable to drug use and exhibit compulsive-like use of cocaine, while others exhibit resilience similar to human subjects. Additionally, the cocaine biobank [28] maintained at the University of California, San Diego (UCSD) completely characterizes the addiction-related behaviors of HS rats and computes an addiction index, permitting the correlation of transcriptomic changes with rat addiction phenotypes. Studies using experimenter-administered, non-contingent cocaine in rodents have demonstrated coding and non-coding RNA expression changes at different stages of cocaine withdrawal in the prefrontal cortex (PFC) and nucleus accumbens (NAc) [29,30,31]. However, abstinence-induced small noncoding RNA (e.g., miRNA) expression changes using a contingent, escalated cocaine intake paradigm merit further investigation.

The present study explored cocaine withdrawal/abstinence-associated miRNA transcriptomic changes in brain reward regions using the HS rat. Extended access to cocaine was followed by two different durations of forced withdrawal and abstinence (18 h and 4 weeks, respectively). miRNA transcriptome profiling was carried out in the cortico-accumbal pathway, including the nucleus accumbens (NAc), a key brain region mediating a variety of behaviors, including reinforcement learning, as well as the pre- and infra-limbic areas of the prefrontal cortex (PFC), which normally exerts executive control of behavior.

## 2. Materials and Methods

### 2.1. Rat Brain Samples

Three groups of snap-frozen HS rat brains were obtained from the Cocaine Biobank located in the Department of Psychiatry at the University of California, San Diego (UCSD) La Jolla, CA 92093, USA. Abstinence rats (4 male rats) had an escalated cocaine self-administration (SA) followed by a 4-week protracted abstinence; withdrawal rats (4 male rats) had an escalated cocaine SA followed by an 18-h withdrawal; and naïve rats (4 male rats) were naïve animals. The rats in the three groups were age-matched at the start of self-administration [the mean age (days) of rats for the 4-week abstinence experiment: 61 ± 9 (mean ± SD); the mean age (days) of rats for the 18-h withdrawal experiment: 64 ± 8 (mean ± SD); and the mean age (days) of naïve rats for the control experiment (naïve: 63 ± 8 (mean ± SD)]. The *t*-tests did not show significant age differences between the three groups of rats at the start of cocaine administration (abstinence vs. withdrawal: t = 0.47, *p* = 0.656; abstinence vs. naïve: t = 0.252, *p* = 0.809; withdrawal vs. naïve: t = 0.22, *p* = 0.829). Detailed information on cocaine SA and withdrawal, as well as behavioral procedures, can be found in a published article [28] and the George Lab protocol (https://www.protocols.io/workspaces/george-lab) (last two visits-November 2022 and December 2022). All procedures were performed following the Guide for the Care and Use of Laboratory Animals from the National Institutes of Health (NIH), and the protocol (IACUC S1906—accessed on 23 February 2023) was approved by the Institutional Animal Care and Use Committees (IACUC) of The Scripps Research Institute and UCSD. Our studies were carried out blinded to the treatment conditions of the rats.

Briefly, rats were anesthetized with vaporized isoflurane (1–5%) and then underwent surgical insertion of an intravenous catheter into the right jugular vein. After surgery and recovery, rats were trained on a fixed ratio 1 (FR1) schedule for self-administering (SA) cocaine (0.5 mg/kg) for 10 sessions, with each session lasting for 2 h. Next, rats were given longer access (LgA) to SA cocaine (0.5 mg/kg) for 14 sessions, with each session lasting for 6 h. Following the last SA session, a subset of rats underwent a protracted period (4 weeks) of forced abstinence and were then euthanized (abstinence: 4 male rats). Another subset of rats were euthanized 18 h following the last SA session (withdrawal: 4 male rats). Finally, a third group of aged-matched naïve rats (naïve: 4 male rats) was used in parallel. All brains were extracted and submerged in a slurry of 2-methylbutane with dry ice (at −30 °C) until fully frozen. Snap-frozen rat brains were then shipped on dry ice to the Boston University School of Medicine. In addition, abstinence and withdrawal rats were subjected to a battery of behavioral tests to compute z-scores for specific addiction-like behaviors, as described previously [28]. These z-scores included the escalation index (Z ESC), the progressive ratio index (motivation or Z PR), and the shock index (compulsivity-like behavior or Z Shock). The overall addiction index was then obtained by averaging relevant behavioral indices. Boston University’s investigators were blinded to whether the rats were naïve or cocaine-experienced rats, and whether they underwent withdrawal or protracted abstinence from cocaine. 

### 2.2. Rat Brain Dissection and Small RNA Sequencing (sRNA-Seq)

Thick sections from snap-frozen rat brains were sliced using a razor blade and a Thermo Scientific HM525 NX Cryostat (Thermo Fisher Scientific, Waltham, MA, USA). The nucleus accumbens (NAc) and infra- and pre-limbic prefrontal cortices (IL and PL) were dissected (Appendix A). The microdissection was carried out rapidly from thick frozen sections placed on the surface of a clean, RNase-free glass platform cooled to −30 °C on dry ice. Anatomical features were used to identify the brain nuclei prior to dissection of specific limbic nuclei.

Total RNAs were isolated from 10 to 30 mg of selected brain nuclei of the limbic circuit using the miRNeasy Mini kit (Qiagen, Valencia, CA, USA). RNA integrity numbers (RIN) and concentrations of 36 total RNA samples [extracted from three brain regions (IL, PL, and NAc) of three groups of rats (abstinence: 4 male rats; withdrawal: 4 male rats; and naïve: 4 male rats)] were measured using the Agilent 2100 Bioanalyzer and Agilent RNA 6000 Nano Kit (Agilent Technologies, Santa Clara, CA, USA) (Appendix A). Small noncoding miRNA transcriptomes in three brain regions (IL, PL, and NAc) of the three groups of rats were profiled using small RNA sequencing (sRNA-seq), as we previously described [21,32]. In short, 250 g of total RNA was used to generate small RNA-seq libraries with the use of the NEBNext Multiplex Small RNA Library Prep Set for Illumina (Set 1) (NEB, Ipswich, MA, USA). cDNA libraries were purified and pooled in an equimolar ratio (12 libraries per pool). They were then sequenced by single-ended 75-bp RNA sequencing on an Illumina HiSeq 2500 Sequencing System (Illumina, CA, USA).

The workflow for the Comprehensive Analysis Pipeline for miRNA Sequencing Data (CAP-miRseq) [33] was used for raw reads (in fastq files) pre-processing, alignment (Reference Sequences: the Rat Genome Assembly rn6), and mature/precursor/novel miRNA qualification and prediction. The read counts of 723 rat mature miRNAs were obtained. The average number of reads per sample was 20,559,895, and the average mapping rate (aligned reads/reads sent to Aligner) was 72.9%. The sRNA-seq fastq files and normalized read counts can be downloaded from the NCBI Gene Expression Omnibus (GEO) database (accession number: GSE212651).

### 2.3. Statistical Analysis

miRNA transcriptomic changes in each of the three brain regions (IL, PL, and NAc) due to a 4-week abstinence or an 18-h withdrawal after escalated cocaine use were analyzed [abstinence (a 4-week abstinence) vs. naïve, as well as withdrawal (an 18-h withdrawal) vs. naïve] following the method applied in our recent study [32]. We also compared miRNA transcriptomes between rats with protracted abstinence and acute withdrawal (i.e., a 4-week abstinence vs. an 18-h withdrawal).

First, the filterByExpr function in edgeR [34] was used to filter out lowly expressed miRNAs, which were not expressed at a biologically meaningful level. Second, miRNA expression levels were normalized using the weighted trimmed mean of M-value (TMM) method through the calcNormFactors function in edgeR [35], which computed scaling factors to convert observed library sizes into effective library sizes. Third, the expression counts were converted into normalized log_2_CPM counts using the function voom [36], by which the mean–variance relationship of the log-counts was analyzed and a precision weight for each miRNA was calculated. Fourth, the lmfit function in Limma [37] was used to fit a linear regression model using weighted least squares for miRNAs. Estimated coefficients and standard errors for a given set of contrasts between groups were computed using the contrasts.fit function, which is also part of the lima package [37]. Finally, a t-statistic test was used to identify significant miRNAs. The RNA integrity number (RIN) of RNA samples was considered a covariate in the differential expression analysis. The decideTests function [38] for multiple testing across genes and contrasts was used to classify miRNAs as significantly negative, not significant, or significantly positive. The differential expression results were visualized by volcano plotting [39]. In addition, the correlation of normalized expression levels of differentially expressed miRNAs [fold-change (FC) ≥ 2.0 and *p* < 0.05] with the addiction index, as well as z-scores (Z ESC: the z-score of escalation of cocaine use; Z PR: the z-score of progressive ratio or motivation to seek cocaine; and Z Shock: the z-score of contingent foot shock or resistance to punishment) was analyzed by a partial correlation analysis [40] with “Group” as a confounding variable.

### 2.4. Bioinformatics Analysis

The function of cocaine withdrawal-associated miRNAs [|fold-change (FC)| ≥ 1.5 and *p* < 0.05], identified by comparing miRNA transcriptomes among the three groups of rats (acute withdrawal vs. naïve, protracted abstinence vs. naïve, and acute withdrawal vs. protracted abstinence), was analyzed using web-server mirPath v.3 (Volos, Greece) [41], which utilizes the DIANA-microT-CDS algorithm to predict miRNA targets (or mRNAs). mirPath v.3 was also used to annotate KEGG (Kyoto Encyclopedia of Genes and Genomes) pathways overrepresented by mRNAs predicted to be targeted by differentially expressed miRNAs.

In addition, the predicted miRNA–mRNA pairs and their associated canonical pathways were analyzed using the miRNA Target Filter function in the Ingenuity Pathway Analysis (IPA, Ingenuity Systems, http://www.ingenuity.com) URL accessed on 29 July 2022. miRNAs that were differentially expressed between groups (|FC| ≥ 1.5 and *p* < 0.05) were run through the miRNA Target pipeline in the IPA to identify their target mRNAs with high confidence or experimentally validated. Next, target mRNAs were filtered to retain those belonging to specific classes of molecules, such as “nuclear receptor” or “transcription factor”. miRNAs were filtered to include those predicted as target genes (or mRNAs) participating in canonical pathways associated with nervous system processes or signaling pathways. Finally, miRNAs and their target mRNAs (after filtering) along with relevant canonical pathways enriched in miRNA target mRNA sets (as determined by the IPA database) were visualized.

## 3. Results

### 3.1. Brain Region-Specific miRNA Expression Changes Due to Acute Withdrawal or Protracted Abstinence following Cocaine Self-Administration (SA)

After normalizing miRNA expression levels and removing low-expression miRNAs, 377 mature miRNAs remained for differential expression analysis. Differentially expressed miRNAs in three brain regions (IL, PL, and NAc) of rats with acute withdrawal (18 h) are visualized using volcano plots (Figure 1). An 18-h acute withdrawal (compared to naïve rats) resulted in differential expression (|FC| ≥ 1.5 and *p* < 0.05) of 21 miRNAs in the IL, 18 miRNAs in the PL, and two miRNAs in the NAc (Table 1). rno-miR-338-3p showed differential expression in both PL and NAc; however, it was upregulated in the PL (log_2_FC = 0.80 or FC = 1.74 and *p* = 0.046) but downregulated in the NAc (log_2_FC = −0.84 or FC = −1.78 and *p* = 0.038). rno-miR-382-3p was downregulated in both IL (log_2_FC = −1.29 or FC = −2.45 and *p* = 0.022) and NAc (log_2_FC = −1.08 or FC = −2.11 and *p* = 0.049). Moreover, differentially expressed miRNAs in three brain regions (IL, PL, and NAc) of rats with a protracted (4-week) protracted abstinence are visualized with volcano plots (Figure 2). A 4-week protracted abstinence (compared to naïve rats) led to differential expression (|FC| ≥ 1.5 and *p* < 0.05) of 23 miRNAs in the IL, seven miRNAs in the PL, and five miRNAs in the NAc (Table 2). Two miRNAs were upregulated in both the IL (rno-miR-935: log_2_FC = 0.81 or FC = 1.75 and *p* = 0.002; rno-miR-770-5p: log_2_FC = 0.83 or FC = 1.77 and *p* = 0.017) and the NAc (rno-miR-935: log_2_FC = 0.70 or FC = 1.63 and *p* = 0.005; rno-miR-770-5p: log_2_FC = 0.66 or FC = 1.58 and *p* = 0.047). Additionally, differential expression (|FC| ≥ 1.5 and *p* < 0.05) of 16 miRNAs in the IL, two miRNAs in the PL, and one miRNA in the NAc were observed when comparing miRNA transcriptome profiles between rats with a 4-week abstinence and an 18-h withdrawal after cessation of cocaine self-administration (Figure 3 and Table 3).

### 3.2. Function of Withdrawal-Associated Brain miRNAs Predicted by DIANA-mirPath

The function of differentially expressed (|FC| ≥ 1.5 and *p* <0.05) miRNAs associated with cocaine withdrawal was annotated by the miRNA pathway analysis web server DIANA-mirPath and presented using bubble plots (Figure 4, Figure 5 and Figure 6). mRNAs predicted to be targeted by 39 miRNAs (Table 1) differentially expressed (|FC| ≥ 1.5 and *p* <0.05) in the IL (21 miRNAs), the PL (18 miRNAs), or the NAc (two miRNAs) due to an 18-h acute withdrawal (withdrawal vs. naïve) were enriched in KEGG pathways, including *gap junctions* (*p* = 0.005; 17 miRNAs:23 mRNAs), *neurotrophin signaling* (*p* = 0.005; 21 miRNAs:36 mRNAs), *MAPK signaling* (*p* = 0.009; 27 miRNAs:58 mRNAs), *hippo signaling* (*p* = 0.016; 22 miRNAs:37 mRNAs), *Wnt signaling* (*p* = 0.026; 26 miRNAs:39 mRNAs), *Ras signaling* (*p* = 0.031; 24 miRNAs:48 mRNAs), and *cocaine addiction* (*p* = 0.043; 10 miRNAs:11 mRNAs) (Figure 4).

Moreover, mRNAs predicted to be targeted by 33 differentially expressed (|FC| ≥ 1.5 and *p* < 0.05) miRNAs (Table 2) in the IL (23 miRNAs), the PL (seven miRNAs), or the NAc (five miRNAs) due to a 4-week abstinence (abstinence vs. naïve) were enriched in KEGG pathways, including *gap junctions* (*p* = 0.0002; 15 miRNAs:21 mRNAs), *cocaine addiction* (*p* = 0.0004; 10 miRNAs:11 mRNAs), *MAPK signaling* (*p* = 0.002; 20 miRNAs:48 mRNAs), *glutamatergic synapses* (*p* = 0.003; 14 miRNAs:24 mRNAs), *morphine addiction* (*p* = 0.004; 10 miRNAs:17 mRNAs), *amphetamine addiction* (*p* = 0.007; 12 miRNAs:16 mRNAs), and *neurotrophin signaling* (*p* = 0.025; 15 miRNAs:26 mRNAs) (Figure 5).

Additionally, 19 miRNAs (including 16 miRNAs in the IL, two miRNAs in the PL, and 1 miRNA in the NAc) (Table 3) with differential expression (|FC| ≥ 1.5 and *p* < 0.05) between rats (acute 18-h withdrawal vs. 4-week protracted abstinence) were predicted to target mRNAs enriched in KEGG pathways, including *cocaine addiction* (*p* = 0.0004; eight miRNAs:10 mRNA), *gap junctions* (*p* = 0.0004; nine miRNAs:18 mRNA), *MAPK signaling* (*p* = 0.0007; 12 miRNAs:46 mRNAs), *synaptic vesicle cycle* (*p* = 0.002; eight miRNAs:12 mRNAs), *glutamatergic synapses* (*p* = 0.013; nine miRNAs:20 mRNAs), *Wnt signaling* (*p* = 0.020; 12 miRNAs:27 mRNAs), *amphetamine addiction* (*p* = 0.021; 10 miRNAs:12 mRNAs), *neurotrophin signaling* (*p* = 0.026; 11 miRNAs:24 mRNAs), *Ras signaling* (*p* = 0.028; 12 miRNAs:35 mRNAs), *morphine addiction* (*p* = 0.028; eight miRNAs:14 mRNAs), and *axon guidance* (*p* = 0.039; 11 miRNAs:23 mRNAs) (Figure 6). Of interest, all three sets of differentially expressed miRNAs potentially regulate the expression of mRNAs involved in the cocaine addiction pathway. Thus, the cocaine addiction pathway was influenced by both acute withdrawal and protracted abstinence from cocaine intake.

### 3.3. Cocaine Withdrawal or Abstinence-Associated miRNA−mRNA-Pathway Network Predicted by IPA

IPA was used to gain further insights into the biological significance of differentially expressed miRNAs and their putative mRNA targets. The numbers of differentially expressed (|FC| ≥ 1.5 and *p* < 0.05) miRNAs identified in each of the three limbic nuclei (IL, PL, and NAc) through the three comparisons (an 18-h acute withdrawal vs. naïve; a 4-week protracted abstinence vs. naïve; and acute withdrawal vs. protracted abstinence) were visualized using Venn diagrams (Figure 7). These miRNAs were associated with acute withdrawal, protracted abstinence, or both.

IPA pathway analysis predicted canonical pathways for all three groups of DE miRNA (withdrawal only, withdrawal and protracted abstinence, and uniquely differentially expressed during protracted abstinence, depicted in the Venn diagram). Additionally, IPA analysis provided predicted targets. We focus on ligand-gated receptors, transcription factors, and signaling pathways. Only targets and pathways that were classified as “experimentally validated and with high confidence” in the IPA database were used to obtain insight into the possible functional role of miRNAs.

All differentially expressed miRNAs in the three groups (WD, abstinence, and both in WD and abstinence) have ligand-gated nuclear receptors and transcription factors as predicted targets. These miRNAs are also predicted to be involved in highly relevant canonical signaling pathways. Canonical signaling pathways identified by IPA analysis include *Glutamate Receptor signaling, Long Term Potentiation (LTP) and Long Term Depression (LTD), and GABA Receptor signaling; CREB signaling in Neurons; Endocannabinoid Developing Neuron Pathway; Endocannabioid Neuronal Synapse pathway; Protein Kinase A signaling and Glucocorticoid Receptor signaling; Dopamine-DARP32 feedback in cAMP signaling; and Nervous system pathways- Endocannabinoid, Serotonin, and Dopamine.*

Restricting miRNAs with target mRNAs involved in nervous system processes or neurotransmitter signaling yielded three miRNAs (rno-miR101a-3p, rno-miR192-3p, and rno-miR137-3p). Pathways predicted to be regulated by these three miRNAs include the *Endocannabinioid Neuronal Synapse Pathway*, *Neuroinflammation signaling Pathway*, *PI3/Akt Signaling*, and *Apoptosis Signaling* for rno-miR101-3p; *DNA Methylation and Transcriptional Repression Signaling Pathway* and *Myelination Signaling Pathway* for rno-miR192-3p; and *Aryl Hydrocarbon Receptor Signaling and Senescence Pathway* for miR137-3p (Appendix A).

IL-PFC: number of differential IL-PFC miRNA expression profiles by three comparisons; PL-PFC: number of differential PL-PFC miRNA expression profiles by three comparisons; and NAc: number of differential NAc miRNA expression profiles by three comparisons. These miRNAs were associated with acute withdrawal, protracted abstinence, or both.

### 3.4. Addiction-like Behaviors and Their Correlation with Expression Levels of Differentially Expressed miRNAs

The three z-scores (Z ESC, Z PR, and Z Shock) and the overall addiction index (based on the three z-scores) of four rats in abstinence (4-week abstinence) and four rats in withdrawal (18-h withdrawal) are summarized in Appendix A. There was no significant difference in the addiction index between abstinence and withdrawal rats (t = 0.25, *p* = 0.808).

We further analyzed the correlation of the addiction index with expression levels of the top differentially expressed miRNAs (|FC| ≥ 2.0 and *p* < 0.05) that were identified by comparing brain transcriptome profiles of the three groups of rats (abstinence vs. naïve, withdrawal vs. naïve, and abstinence vs. withdrawal). Four miRNAs (rno-miR-582-3p, rno-miR-292-5p, rno-miR-551b-3p, and rno-miR-382-3p) in the IL, three miRNAs (rno-miR-381-3p, rno-miR-1188-3p, and rno-miR-872-5p) in the PL, and one miRNA (rno-miR-382-3p) in the NAc showed differential expression (|FC| ≥ 2.0 and *p* < 0.05) in rats with escalated cocaine use followed by an 18-h withdrawal (withdrawal vs. naïve) (Table 1). Moreover, four miRNAs (rno-miR-101a-3p, rno-miR-666-3p, rno-miR-136-3p, and rno-miR-192-5p) in the IL showed differential expression (|FC| ≥ 2.0 and *p* < 0.05) in rats with escalated cocaine use followed by a 4-week protracted abstinence (abstinence vs. naïve) (Table 2). Additionally, nine miRNAs (rno-miR-187-3p, rno-miR-448-3p, rno-miR-382-3p, rno-miR-101a-3p, rno-miR-137-3p, rno-miR-499-5p, rno-miR-551b-3p, rno-miR-764-5p, and rno-miR-292-5p) in the IL and one miRNA (rno-miR-1188-3p) in the PL showed differential expression (|FC| ≥ 2.0 and *p* < 0.05) between rats with a 4-week abstinence and an 18-h withdrawal (abstinence vs. withdrawal) (Table 3). The correlation of the normalized expression levels of these differentially expressed miRNAs noted above in three brain regions (IL, PL, and NAc) with the addiction index in abstinence and withdrawal rats together was analyzed by a partial correlation analysis with “Group” as the confounding variable. No significant correlation of expression levels of the above miRNAs in each of the three brain regions (IL, PL, and NAc) with the addiction index was observed (Appendix A). However, the expression level of rno-miR-381-3p in the IL was significantly correlated with Z ESC (*P*_corr_ = 0.021) (Appendix A). The expression levels of one miRNA in the IL and seven miRNAs in the NAc were significantly correlated with Z PR (IL rno-miR-448-3p: *P*_corr_ = 0.009; NAc rno-miR-101a-3p: *P*_corr_ = 0.018; NAc rno-miR-137-3p: *P*_corr_ = 0.004; NAc rno-miR-192-5p: *P*_corr_ = 0.007; NAc rno-miR-381-3p: *P*_corr_ = 0.040; NAc rno-miR-448-3p: *P*_corr_ = 0.031; NAc rno-miR-582-3p: *P*_corr_ = 0.049; and NAc rno-miR-872-5p: *P*_corr_ = 0.004) (Appendix A). Additionally, there was no significant correlation between Z Shock and the expression levels of miRNAs in the three brain regions (Appendix A).

## 4. Discussion

Substance use disorders (SUDs) are characterized by an overpowering loss of control over drug seeking and a compulsive intake of drugs despite adverse consequences [42]. The transition from casual or recreational drug use to compulsive use happens only in a small fraction of human subjects and animal models, but the mechanisms underpinning this transition are unclear [43,44,45,46]. These observations highlight the variability among individuals in the degree of vulnerability for compulsive drug use [47,48] and underscore the importance of using a genetically diverse rat strain (similar to the heterogeneity of human populations) to conduct research. Here we report on miRNA transcriptome alterations in HS rats during withdrawal and prolonged abstinence and compared with drug-naïve rats as a means of identifying changes relevant to the increased propensity for drug use and/or resilience.

Much is known regarding chronic drug use-induced neuroplasticity at the cellular and molecular levels in the case of cocaine in a region-specific manner in the brain [49,50]. In contrast, circuit-wide transcriptomic changes induced by chronic drug use followed by prolonged abstinence are less well understood. Mounting evidence shows that repeated misuse of drugs reprograms the genome throughout the cortico-accumbal circuit after prolonged abstinence [31,51] and may be key in causing vulnerability or resilience to chronic drug use. We have examined cocaine withdrawal- and abstinence-associated miRNA expression changes in the infralimbic (IL) and prelimbic (PL) subdivisions of the medial prefrontal cortex (mPFC), as well as the nucleus accumbens (NAc). Alterations in the expression of miRNAs in these limbic nuclei are of interest since the two subregions of the mPFC drive opposing drug use-related behaviors. The PL drives drug seeking, while the IL inhibits these behaviors [52,53]. Vulnerability to drug seeking increases in correlation with a long duration of abstinence. This phenomenon of incubation of cocaine craving [54,55,56], which is an increased responsivity to cocaine and withdrawal-associated cues, is seen not only in preclinical models but also in humans [57]. In human subjects and animal models, negative emotional states increase during withdrawal from drug use, and resumption of drug taking may alleviate these aversive states and increase relapse vulnerability [58,59,60,61].

Not surprisingly, miRNAs, by virtue of their ability for posttranscriptional regulation of mRNAs and hence protein synthesis, can directly affect cellular functions. Several studies have demonstrated a direct effect of miRNAs on modulating behavioral patterns of drug abuse-related behavior. Overexpression or knockdown of specific miRNAs in vivo in specific limbic nuclei influences addiction behaviors. For instance, motivation for cocaine [62] and vulnerability to cocaine use disorder [12,63] can be regulated by miRNAs. Viral vector-mediated overexpression of miR-9 in the NAc increases the self-administration of oxycodone in rats [64]. The direct effect of miRNAs on behavioral phenotypes indicates the importance of the miRNA transcriptome in determining an individual’s vulnerability to drug addiction and the need for studying miRNA alterations in SUDs.

As an initial step in unraveling the critical functional role of miRNAs expressed in the limbic regions (IL, PL, and NAc) for regulating behaviors relevant to cocaine use disorder (CUD), we have investigated miRNA transcriptomic changes associated with acute withdrawal and protracted abstinence following escalated cocaine. The number of miRNAs that were selectively differentially expressed in specific time intervals, namely withdrawal or abstinence, or those that were differentially expressed persistently in both abstinence and withdrawal in the three brain regions were visualized by a Venn diagram (Figure 7). Importantly, the use of these three groups of rats enabled the identification of miRNAs that are differentially expressed and are unique to specific periods of withdrawal or abstinence (Figure 7) and has permitted insight into the possible functional roles of specific miRNAs in promoting craving or resilience in each of these abstinence periods.

To gain an understanding of mechanisms underlying drug use-related behaviors, bioinformatics platforms and tools were leveraged to predict canonical pathways regulated by miRNAs. Pathway identification provides insight into possible signaling pathways and mechanisms underlying drug use-relevant behaviors. The Diana-mirPath/KEGG Pathway analysis reveals that multiple addiction-relevant signaling pathways (including *cocaine*, *morphine*, and *amphetamine addictions*) are over-represented by genes targeted by miRNAs identified in the three limbic regions (Figure 4, Figure 5 and Figure 6). KEGG pathways overrepresented by genes potentially targeted by miRNAs associated with both withdrawal and abstinence include *cocaine addiction*, *neurotrophin signaling*, *MAPK signaling*, and *gap junctions* (Figure 4 and Figure 5). Ample evidence suggests that neurotrophins and the MAPK signal transduction cascade play important roles in behavioral sensitization to cocaine [65]. Studies have also shown that cocaine exposure impacts gap junctional communication between neurons [66]. The increase in the vasopressin pathway during withdrawal is significant since it promotes stress-related substance use behavior [67], whereas oxytocin, observed to increase during abstinence, is an anti-stress factor [68,69]. The FOXO family of transcription factors elicits the enhancement of SUDS-related behaviors [70].

Thus far, very few transcription factors other than members of the Fos family have been identified in drug abuse studies [71]. However, recent studies [31,51,72] support a prominent role of transcription factors and nuclear receptor families in the addictive process. Consistent with these recent studies, the IPA pathway analysis of our miRNA seq data using only experimentally validated and high confidence targets reveals that predicted targets of DE miRNAs in IL and PL during WD, abstinence alone, and both in WD and abstinence are enriched in CREB and other transcription factors (Appendix A). This is of functional significance since transcription factors are critical upstream modulators of cocaine-induced plasticity [73].

It is known that dysregulation of endogenous plasticity mechanisms in the limbic circuit is likely to promote SUDs. For instance, in the NAc, cascades of homeostatic plasticity from bi-directional synaptic-membrane crosstalk occur and are critically involved in cocaine-induced plasticity and consequent cocaine craving [74,75], and there is a persistent strengthening of the PFC−NAc pathway [76]. Preventing dysregulation of homeostatic plasticity and restoring normal plasticity is likely to provide resilience. The IPA bioinformatic analysis using only experimentally validated and high confidence targets predict that DE miRNAs identified in the present study target many relevant plasticity pathways. These include the glutamate and GABA receptor pathways, LTP and LTD, the glucocorticoid signaling pathway, and the endocannabinoid signaling pathway. Predicted targets are interesting in that they include, for example, NMDA receptor subunits, AMPA receptor auxillary proteins, and glutamate transporters—proteins that are crucial for eliciting neuronal plasticity (Appendix A).

Additionally, even further restriction of identified miRNA candidates to only those with target mRNAs involved in nervous system processes or neurotransmitter signaling yielded three miRNAs (rno-miR101a-3p, rno-miR192-3p, and rno-miR137-3p). Pathways predicted to be regulated by these three miRNAs include the *Endocannabinioid Neuronal Synapse Pathway*, *Neuroinflammation signaling Pathway*, *PI3/Akt Signaling*, and *Apoptosis Signaling* for rno-miR101-3p; *DNA Methylation and Transcriptional Repression Signaling Pathway* and *Myelination Signaling Pathway* for rno-miR192-3p; and *Aryl Hydrocarbon Receptor Signaling and Senescence Pathway* for miR137-3p (Appendix A). Once again, these pathways underscore the functional significance of the DE miRNA identified in our study and highlight the importance of plasticity, neuroinflammation, and epigenetic processes such as DNA methylation in CUD.

Recently, Domingo-Rodriguez and colleagues, using cocaine self-administration in mice, have shown that miRNAs regulate vulnerability vs. resilience, thus contributing to interindividual variability [63]. Consistent with this study, we find that specific miRNAs were significantly correlated with addiction-relevant behaviors, such as escalated cocaine intake (i.e., correlation with the z-score for escalation or Z ESC) and motivation to take cocaine (i.e., correlation with the z-score for progressive ratio or Z PR) (Appendix A). Of interest, the expression levels of rno-miR-101a-3p and rno-miR-137-3p in the NAc were significantly correlated with Z PR (Appendix A). This finding implies that the altered expression of these two miRNAs in NAc could influence cocaine use behavior by changing the rats’ motivation to seek cocaine. High motivation for drugs in rats, as measured by the maximal effort exerted for drug infusions, may mimic attributes seen in human subjects with SUDs.

Limitations of the study include a small sample size. Another is the lack of a larger number of differentially expressed miRNAs identified in the NAc. This may be due to extracting RNA from the entire NAc and not separating the shell from the core subdivisions. The shell and core subdivisions of the NAc have different roles in addiction [77]. Only male rats have been used in this study, but female rats will be tested in our future studies to elucidate sex differences since these are known to occur in impulsive and compulsive behaviors such as addiction [78].

## 5. Conclusions

This work is an important, initial step in identifying miRNA-mediated regulation of plasticity induced by protracted abstinence from escalated cocaine intake. We have identified rat brain limbic region miRNAs that are likely to modulate behaviors that are relevant to cocaine abuse. By identifying escalated cocaine self-administration and withdrawal/abstinence-induced miRNAs, we have uncovered an additional layer of epigenetic modulation of CUD. Increased understanding of epigenome remodeling, brain plasticity, and vulnerability/resilience to drug abuse will provide insights for developing novel, effective therapeutic strategies for SUDs, including CUD.

## Figures and Tables

**Figure 1 biomedicines-11-01368-f001:**
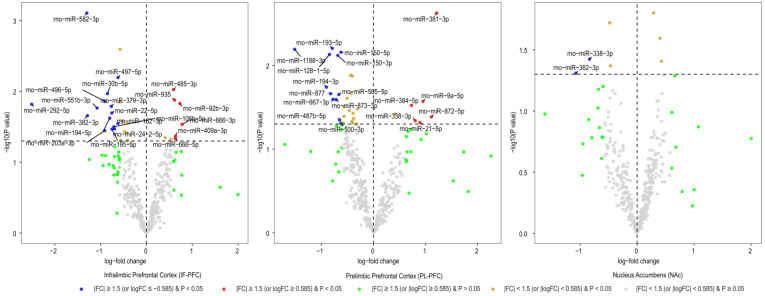
Volcano plotting of differentially expressed rat brain miRNAs due to an 18-h cocaine withdrawal (an 18-h cocaine withdrawal vs. naïve rats). Scattered points represent miRNAs. The x-axis is the log_2_ fold-change (FC) of miRNA expression levels between the two groups of rats, and the y-axis is the *p*-value based on −log10.

**Figure 2 biomedicines-11-01368-f002:**
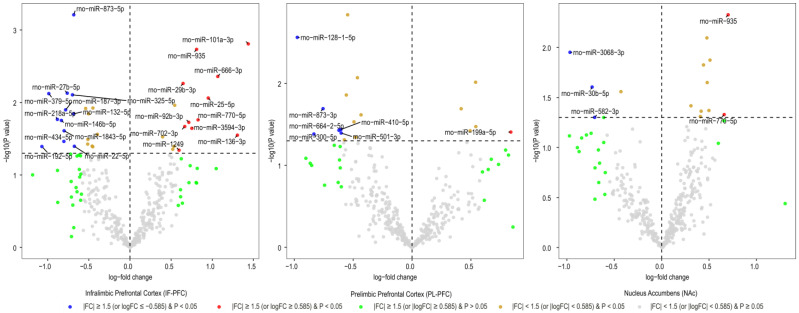
Volcano plotting of differentially expressed rat brain miRNAs due to a 4-week cocaine abstinence (a 4-week cocaine abstinence vs. naïve rats). Scattered points represent miRNAs. The x-axis is the log_2_ fold-change (Fc) of miRNA expression levels between the two groups of rats, and the y-axis is the *p*-value based on −log10.

**Figure 3 biomedicines-11-01368-f003:**
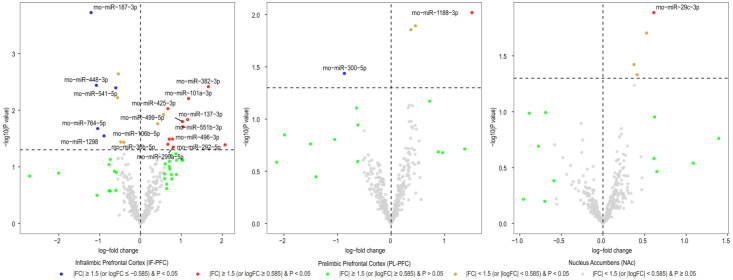
Volcano plotting of miRNAs with differential expression between rats with two stages of cocaine withdrawal (18-h cocaine withdrawal vs. 4-week cocaine abstinence). Scattered points represent miRNAs. The x-axis is the log_2_ fold-change (Fc) of miRNA expression levels between the two groups of rats, and the y-axis is the *p*-value based on −log10.

**Figure 4 biomedicines-11-01368-f004:**
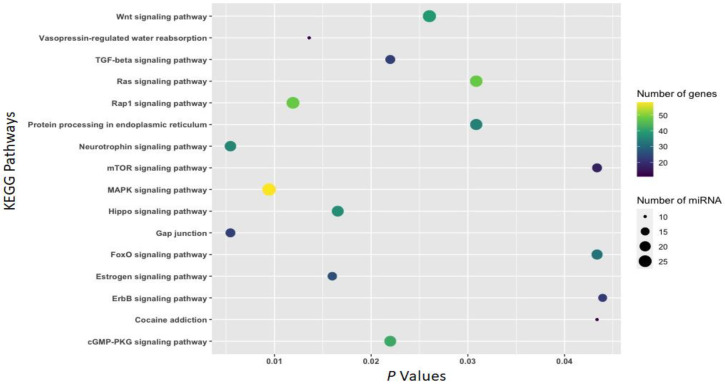
KEGG pathways overrepresented by mRNAs potentially targeted by 39 miRNAs (Table 1) associated with an 18-h cocaine withdrawal. All 39 miRNAs were associated with an 18-h cocaine withdrawal in either of the three brain regions (IL, PL, and NAc). The mirPath v.3 online tool (https://dianalab.e-ce.uth.gr/html/mirpathv3/index.php?r=mirpath; accessed on 27 January 2023) was used to annotate KEGG pathways overrepresented by mRNAs predicted to be targeted by these miRNAs.

**Figure 5 biomedicines-11-01368-f005:**
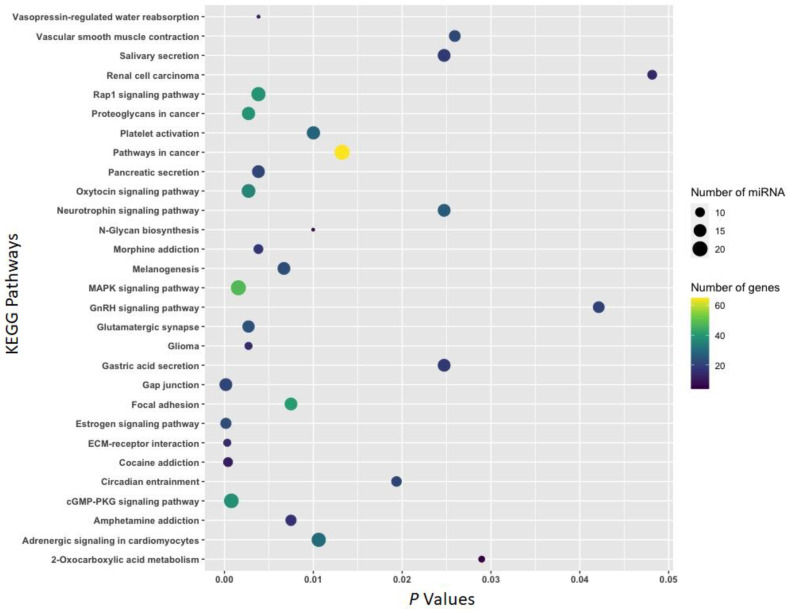
KEGG pathways overrepresented by mRNAs potentially targeted by 33 miRNAs (Table 2) associated with a 4-week cocaine abstinence. All 33 miRNAs were associated with a 4-week cocaine abstinence in either of the three brain regions (IL, PL, and NAc). The mirPath v.3 online tool was used to annotate KEGG pathways overrepresented by mRNAs predicted to be targeted by these miRNAs.

**Figure 6 biomedicines-11-01368-f006:**
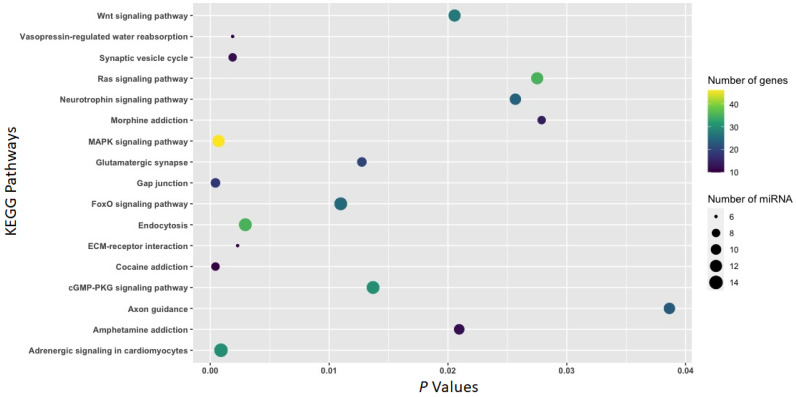
KEGG pathways overrepresented by mRNAs potentially targeted by miRNAs (Table 3) with differential expression between two stages (18-h vs. 4-week) of cocaine withdrawal rats. Here, 19 miRNAs showed differential expression between two stages (18-h vs. 4-week) of cocaine withdrawal for rats in either of the three brain regions (IL, PL, and NAc). The mirPath v.3 online tool was used to annotate KEGG pathways overrepresented by mRNAs predicted to be targeted by these miRNAs.

**Figure 7 biomedicines-11-01368-f007:**
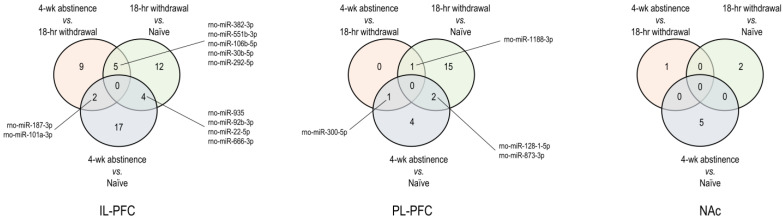
Venn diagram plotting of the numbers of differentially expressed miRNAs in three rat brain regions due to an 18-h cocaine withdrawal, a 4-week cocaine abstinence, or both.

**Table 1 biomedicines-11-01368-t001:** The miRNA expression changes of the 18-h cocaine withdrawal-induced rat brain (|FC| ≥ 1.5 and *p* < 0.05).

Infralimbic Prefrontal Cortex miRNA	Log_2_FC	AveExpr	t	B	*p*.Value	adj.*p*.Val
rno-miR-582-3p	−1.30	2.98	−3.73	−1.30	0.001	0.292
rno-miR-497-5p	−0.62	4.23	−2.93	−2.42	0.006	0.512
rno-miR-485-3p	0.59	10.51	2.77	−2.60	0.009	0.512
rno-miR-30b-5p	−0.85	3.43	−2.72	−2.84	0.011	0.512
rno-miR-935	0.61	6.30	2.63	−2.87	0.013	0.512
rno-miR-496-5p	−0.91	0.94	−2.61	−3.27	0.014	0.512
rno-miR-92b-3p	0.73	12.03	2.58	−2.93	0.015	0.512
rno-miR-292-5p	−2.51	1.20	−2.57	−3.17	0.015	0.512
rno-miR-379-3p	−0.77	5.63	−2.54	−3.02	0.016	0.512
rno-miR-551b-3p	−1.08	1.07	−2.52	−3.35	0.017	0.512
rno-miR-22-5p	−0.76	6.18	−2.45	−3.17	0.020	0.512
rno-miR-382-3p **	−1.29	2.54	−2.41	−3.34	0.022	0.512
rno-miR-194-5p	−0.80	4.86	−2.38	−3.30	0.024	0.512
rno-miR-152-3p	−0.62	5.86	−2.30	−3.42	0.028	0.512
rno-miR-666-3p	0.78	0.82	2.29	−3.63	0.029	0.512
rno-miR-106b-5p	−0.71	2.09	−2.25	−3.59	0.032	0.512
rno-miR-24-2-5p	−0.70	6.03	−2.22	−3.54	0.034	0.512
rno-miR-195-5p	−0.75	3.92	−2.22	−3.55	0.034	0.512
rno-miR-203a-3p	−0.91	2.37	−2.20	−3.63	0.036	0.512
rno-miR-409a-3p	0.63	12.53	2.11	−3.73	0.043	0.512
rno-miR-666-5p	0.63	5.35	2.08	−3.76	0.046	0.512
**Prelimbic Prefrontal** **Cortex miRNAs**	**Log_2_FC**	**AveExpr**	**t**	**B**	***p*.Value**	**adj.*p*.Val**
rno-miR-381-3p	1.22	1.56	3.32	−4.34	0.002	0.472
rno-miR-193-5p	−0.79	2.45	−2.94	−4.35	0.006	0.472
rno-miR-1188-3p	−1.51	0.97	−2.93	−4.42	0.006	0.472
rno-miR-150-5p	−0.62	9.10	−2.89	−4.09	0.007	0.472
rno-miR-128-1-5p	−0.84	4.04	−2.87	−4.25	0.007	0.472
rno-miR-150-3p	−0.68	4.41	−2.86	−4.23	0.008	0.472
rno-miR-194-3p	−0.90	1.50	−2.50	−4.46	0.018	0.520
rno-miR-877	−0.82	4.76	−2.42	−4.32	0.022	0.520
rno-miR-505-5p	−0.67	4.92	−2.41	−4.32	0.022	0.520
rno-miR-667-3p	−0.77	6.76	−2.35	−4.29	0.025	0.520
rno-miR-873-3p	−0.71	6.71	−2.35	−4.29	0.026	0.520
rno-miR-9a-5p	0.95	14.95	2.32	−4.22	0.027	0.520
rno-miR-384-5p	0.73	6.72	2.28	−4.31	0.030	0.520
rno-miR-872-5p	1.11	4.52	2.13	−4.41	0.041	0.520
rno-miR-487b-5p	−0.65	2.92	−2.09	−4.47	0.045	0.520
rno-miR-338-3p*	0.80	3.54	2.08	−4.44	0.046	0.520
rno-miR-21-5p	0.89	6.17	2.06	−4.38	0.048	0.520
rno-miR-500-3p	−0.63	0.84	−2.05	−4.52	0.049	0.520
**Nucleus Accumbens** **miRNAs**	**Log_2_FC**	**AveExpr**	**t**	**B**	***p*.Value**	**adj.*p*.Val**
rno-miR-338-3p *	−0.84	3.54	−2.17	−4.43	0.038	0.847
rno-miR-382-3p **	−1.08	2.54	−2.05	−4.48	0.049	0.847

Log_2_FC: the log2 fold-change (FC) between cases and controls; AveExpr: average miRNA expression level; t: the t-statistic used to assess differential expression; B: the empirical Bayes log odds of differential expression; *p*.Value: the *p*-value for differential expression (unadjusted for multiple testing); adj.*p*.Val: the *p*-value adjusted for multiple testing (the default is Benjamini−Hochberg). *, **: Differentially expressed miRNAs in multiple brain regions.

**Table 2 biomedicines-11-01368-t002:** The miRNA expression changes of the 4-week cocaine abstinence-induced rat brain (|FC| ≥ 1.5 and *p* < 0.05).

Infralimbic Prefrontal Cortex miRNAs	Log_2_FC	AveExpr	t	B	*p*. Value	adj.*p*.Val
rno-miR-873-5p	−0.68	5.75	−3.81	−0.46	0.001	0.231
rno-miR-101a-3p	1.44	4.99	3.47	−1.23	0.002	0.231
rno-miR-935 *	0.81	6.30	3.41	−1.31	0.002	0.231
rno-miR-666-3p	1.07	0.82	3.08	−2.47	0.004	0.358
rno-miR-29b-3p	0.65	5.37	2.99	−2.17	0.005	0.358
rno-miR-27b-5p	−0.76	4.14	−2.87	−2.44	0.007	0.358
rno-miR-379-5p	−0.99	11.08	−2.86	−2.41	0.008	0.358
rno-miR-325-5p	−0.70	8.24	−2.84	−2.44	0.008	0.358
rno-miR-25-5p	0.95	2.19	2.80	−2.70	0.009	0.358
rno-miR-187-3p	−0.78	7.84	−2.65	−2.82	0.013	0.358
rno-miR-132-5p	−0.68	10.79	−2.60	−2.92	0.014	0.358
rno-miR-218a-5p	−0.88	11.47	−2.52	−3.06	0.017	0.368
rno-miR-770-5p **	0.83	6.89	2.51	−3.06	0.017	0.368
rno-miR-146b-5p	−0.83	10.78	−2.51	−3.08	0.018	0.368
rno-miR-92b-3p	0.71	12.03	2.48	−3.14	0.019	0.370
rno-miR-702-3p	0.67	7.21	2.42	−3.23	0.021	0.401
rno-miR-3594-3p	0.75	3.26	2.40	−3.27	0.023	0.405
rno-miR-1843-5p	−0.80	9.52	−2.36	−3.34	0.025	0.418
rno-miR-136-3p	1.31	1.64	2.30	−3.47	0.028	0.440
rno-miR-434-5p	−0.80	9.85	−2.21	−3.60	0.034	0.462
rno-miR-22-5p	−0.68	6.18	−2.14	−3.70	0.040	0.462
rno-miR-192-5p	−1.07	5.14	−2.14	−3.68	0.041	0.462
rno-miR-1249	0.60	8.07	2.08	−3.81	0.046	0.487
**Prelimbic Prefrontal** **Cortex miRNAs**	**Log_2_FC**	**AveExpr**	**t**	**B**	***p*.Value**	**adj.*p*.Val**
rno-miR-128-1-5p	−0.98	4.04	−3.26	−2.27	0.003	0.513
rno-miR-873-3p	−0.76	6.71	−2.45	−3.26	0.020	0.823
rno-miR-410-5p	−0.60	1.67	−2.19	−3.88	0.036	0.823
rno-miR-664-2-5p	−0.62	6.21	−2.18	−3.63	0.037	0.823
rno-miR-199a-5p	0.84	2.79	2.15	−3.80	0.039	0.823
rno-miR-501-3p	−0.60	4.44	−2.14	−3.71	0.040	0.823
rno-miR-300-5p	−0.83	0.95	−2.13	−3.97	0.041	0.823
**Nucleus Accumbens** **miRNAs**	**Log_2_FC**	**AveExpr**	**t**	**B**	***p*.Value**	**adj.*p*.Val**
rno-miR-935 *	0.70	6.30	3.05	−4.09	0.005	0.853
rno-miR-3068-3p	−0.97	3.13	−2.70	−4.37	0.011	0.853
rno-miR-30b-5p	−0.73	3.43	−2.36	−4.40	0.025	0.853
rno-miR-770-5p **	0.66	6.89	2.07	−4.36	0.047	0.853
rno-miR-582-3p	−0.71	2.98	−2.04	−4.47	0.050	0.853

Log_2_FC: the log2 fold-change (FC) between cases and controls; AveExpr: average miRNA expression level; t: the t-statistic used to assess differential expression; B: the empirical Bayes log odds of differential expression; *p*.Value: the *p*-value for differential expression (unadjusted for multiple testing); adj.*p*.Val: the *p*-value adjusted for multiple testing (the default is Benjamini−Hochberg). *, **: Differentially expressed miRNAs in multiple brain regions.

**Table 3 biomedicines-11-01368-t003:** Rat brain miRNA differential expression (|FC| ≥ 1.5 and *p* < 0.05) between two stages (18-h vs. 4-week) of cocaine withdrawal.

Infralimbic Prefrontal Cortex miRNAs	Log_2_FC	AveExpr	t	B	*p*.Value	adj.*p*.Val
rno-miR-187-3p	−1.21	7.84	−4.24	0.40	0.0002	0.071
rno-miR-448-3p	−1.08	1.90	−3.15	−2.24	0.004	0.302
rno-miR-382-3p	1.66	2.54	3.13	−2.23	0.004	0.302
rno-miR-541-5p	−0.61	11.65	−3.11	−1.92	0.004	0.302
rno-miR-101a-3p	1.17	4.99	2.94	−2.32	0.006	0.333
rno-miR-425-3p	0.67	2.66	2.77	−2.77	0.009	0.439
rno-miR-137-3p	1.15	2.76	2.59	−3.05	0.015	0.540
rno-miR-499-5p	1.03	3.45	2.55	−3.05	0.016	0.540
rno-miR-551b-3p	1.04	1.07	2.46	−3.36	0.020	0.564
rno-miR-764-5p	−1.05	2.51	−2.43	−3.28	0.021	0.565
rno-miR-1298	−0.89	8.11	−2.30	−3.43	0.028	0.710
rno-miR-496-3p	0.78	3.46	2.24	−3.52	0.032	0.714
rno-miR-106b-5p	0.70	2.09	2.24	−3.57	0.032	0.714
rno-miR-30b-5p	0.67	3.43	2.14	−3.66	0.040	0.730
rno-miR-292-5p	2.07	1.20	2.13	−3.71	0.041	0.730
rno-miR-299a-5p	0.80	2.16	2.09	−3.76	0.045	0.763
**Prelimbic Prefrontal** **Cortex miRNAs**	**Log_2_FC**	**AveExpr**	**t**	**B**	***p*.Value**	**adj.*p*.Val**
rno-miR-1188-3p	1.51	0.97	2.76	−4.45	0.010	0.998
rno-miR-300-5p	−0.87	0.95	−2.19	−4.51	0.037	0.998
**Nucleus Accumbens** **miRNAs**	**Log_2_FC**	**AveExpr**	**t**	**B**	***p*.Value**	**adj.*p*.Val**
rno-miR-29c-3p	0.61	3.55	2.64	−4.34	0.013	0.998

Log_2_FC: the log2 fold-change (FC) between cases and controls; AveExpr: average miRNA expression level; t: the t-statistic used to assess differential expression; B: the empirical Bayes log odds of differential expression; *p*.Value: the *p*-value for differential expression (unadjusted for multiple testing); adj.*p*.Val: the *p*-value adjusted for multiple testing (the default is Benjamini−Hochberg).

## Data Availability

The raw data supporting the conclusions of this article can be downloaded from the NCBI Gene Expression Omnibus (GEO) database (Accession Number: GSE212651).

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
