# Peer review of "Abstinence from Escalation of Cocaine Intake Changes the microRNA Landscape in the Cortico-Accumbal Pathway"

_biomedicines, 2023, doi:10.3390/biomedicines11051368_

Round 1
Reviewer 1 Report
The work presented by Kumaresan et al., entitled “Abstinence from Escalation of Cocaine Intake Changes the mi-croRNA Landscape in the Cortico-Accumbal Pathway” affords robust data concerning the involvement of epigenetic neuroadaptations induced by cocaine intake in HS rats, throughout the miRNA transcriptome study and ulterior analysis using DIANA-mirPath and IPA.
From my point of view the article is suitable for publication as soon as authors take into consideration a couple of observations that need to be elucidated.
1. From the methodological point of view:
- RIN values are not provided. Which is the gap found? Are the RIN values > 8.5 at least? Please explain lines 199-200
- sRNA-seq was performed for the 36 samples, right? Why did not you use a pool of the 4 saples for each structure?
2. Authors found many significative changes in miRNA and link them with determinate functions. However, in order to validate these expression modifications, some qPCR need to be done.
Reviewer 2 Report
The study by Kumaresan and colleagues investigated changes in microRNA expression in the cortico-accumbal pathway during acute withdrawal and protracted abstinence following escalated cocaine intake. Small RNA sequencing was used to analyze miRNA transcriptomic changes in the cortico-accumbal pathway of rats with extended access to cocaine self-administration. The study found that acute and protracted abstinence from escalated cocaine intake led to differential expression of miRNAs in the cortico-accumbal pathway, potentially targeting mRNAs enriched in pathways associated with addictive behaviors. Although this paper addresses an important topic and uses relevant methods, there are major limitations that need to be addressed.
1. The search in the GEO database for the accession number GSE212651 does not reveal any matches. Please verify.
2. The authors report that they performed behavioral testing (lines 98-101 and 145-150), but no data is presented. Correlating miRNA expression to behavioral data is essential for this study.
3. Methods are not adequately reported:
a. Microdissection is a misleading term here. Usually, microdissection refers to laser-capture methods. How thick were the sections? What means "anatomical features"? I find it extremely challenging to distinguish those regions in non-stained brain tissue. How did you ensure that the samples obtained from different animals were equal in their size?
b. Line 134-135 "...rats were anesthetized with vaporized isoflurane (1-5%) and then underwent surgical insertion of an intravenous catheter into the right jugular vein". Why was it necessary to introduce the catheter?
c. The described procedure in lines 143-144 hardly qualifies for snap-freezing because it takes minutes.
d. The number of animals is very low in this study (n=4). Is this justified?
4. It is difficult to grab the essential information on the figures. The authors should put more focus on the potential role of the differentially expressed miRNAs in regulating brain function. Some of the reported pathways are not pathways, such as gap junction, addition types, and glioma. The informative figure is number 7, but it remains not clear how those miRNAs are suggested to affect brain function.
5. Performing additional studies of CUD-associated brain remodeling is highly recommended. Without correlating miRNA expression to behavior and brain structure (this can be done by immunohistochemistry, MRI methods, tract tracing, or optogenetics), this study does not provide sufficient advancement compared to previous studies (there are several datasets related to cocaine addiction available: GDS5047, GDS3703, GDS2993, GDS2311, GDS1608, GDS255)
Round 2
Reviewer 1 Report
Article is suitable for publication
Reviewer 2 Report
Although the authors did not perform the desirable experiments, I understand that this can be due to the lack of time and funding. However, I trust that the authors will validate these results in the future and conduct more mechanistic studies.